# Germline de novo mutations in families with Mendelian cancer syndromes caused by defects in DNA repair

Kitty Sherwood [1], Joseph C. Ward[2], Ignacio Soriano [2], Lynn Martin[3], Archie Campbell [4], Raheleh Rahbari [5], Ioannis Kafetzopoulos[1], Duncan Sproul[1], Andrew Green[6], Julian R. Sampson[7], Alan Donaldson[8], Kai-Ren Ong [9], Karl Heinimann[10], Maartje Nielsen [11], Huw Thomas[12], Andrew Latchford[12], Claire Palles [3] ✉ & Ian Tomlinson [2] ✉

DNA repair defects underlie many cancer syndromes. We tested whether de novo germline mutations (DNMs) are increased in families with germline defects in polymerase proofreading or base excision repair. A parent with a single germline *POLE* or *POLD1* mutation, or biallelic *MUTYH* mutations, had 3-4 fold increased DNMs over sex-matched controls. *POLE* had the largest effect. The DNMs carried mutational signatures of the appropriate DNA repair deficiency. No DNM increase occurred in offspring of *MUTYH* heterozygous parents. Parental DNA repair defects caused about 20–150 DNMs per child, additional to the ~60 found in controls, but almost all extra DNMs occurred in non-coding regions. No increase in post-zygotic mutations was detected, excepting a child with bi-allelic *MUTYH* mutations who was excluded from the main analysis; she had received chemotherapy and may have undergone oligoclonal haematopoiesis. Inherited DNA repair defects associated with base pair-level mutations increase DNMs, but phenotypic consequences appear unlikely.

Inherited defects in genome stability or DNA repair can lead to hypermutation and hence to Mendelian cancer predisposition syndromes. In some cases, the phenotype also includes defects in development or features reminiscent of accelerated aging or degeneration[1–3]. The understanding of somatic mutational processes has recently increased appreciably[4–8], and studies have demonstrated variably raised somatic mutation rates in different tissues from DNA repair-deficient patients[9,10]. However, an unresolved issue for these patients is whether germline mutation rates are raised and, if so, the likely clinical impact on their children.

De novo mutations (DNMs) take many forms, ranging from defects at the scale of the DNA base pair to chromosomal aneusomy. DNMs are associated with a wide and complex variety of disease phenotypes, including developmental disorders and very early onset

[1]Cancer Research UK Edinburgh Centre and MRC Human Genetics Unit, Institute of Genomics and Cancer, Crewe Road, Edinburgh EH4 2XU, UK. [2]Dept of Oncology, University of Oxford, Old Road Campus Research Building, Roosevelt Drive, Oxford OX3 7DQ, UK. [3]Institute of Cancer and Genomic Sciences, University of Birmingham Medical School, Vincent Drive, Edgbaston, Birmingham B15 2JJ, UK. [4]Centre for Genetics and Experimental Medicine, Institute of Genetics and Cancer, Western General Hospital, Crewe Road, Edinburgh EH4 2XU, UK. [5]Wellcome Sanger Institute, Wellcome Genome Campus, Hinxton, UK. [6]Department of Clinical Genetics, Children's Health Ireland and School of Medicine University College, Dublin, Ireland. [7]Institute of Medical Genetics, Division of Cancer and Genetics, Cardiff University School of Medicine, Cardiff, UK. [8]Bristol Regional Clinical Genetics Service, St Michael's Hospital, Southwell Street, Bristol BS2 8EG, UK. [9]West Midlands Regional Genetics Service, Birmingham Women's and Children's NHS Foundation Trust, Birmingham, UK. [10]Institute for Medical Genetics and Pathology, University Hospital Basel, Basel, BS, Switzerland. [11]Department of Clinical Genetics, Leiden University Medical Centre, 2333 ZA Leiden, the Netherlands. [12]St Mark's Hospital, Watford Road, Harrow HA1 3UJ, UK. ✉e-mail: c.palles@bham.ac.uk; ian.tomlinson@oncology.ox.ac.uk

cancers[11,12]. There are patchy reports of non-cancer phenotypes consistent with germline DNMs in the offspring of cancer patients with DNA repair defects, but these could represent chance, ascertainment bias, or even early somatic mutations in children who have inherited the DNA repair deficiency. Indeed, it is theoretically possible that DNM burden is not raised at all, owing to protective factors such as selection against mutant haploid gametes and enhanced DNA repair in the germ line. Furthermore, several other factors could confound DNM reporting, including parental exposure to systemic anti-cancer genotoxic therapy and a general tendency to under-report observations of uncertain importance in rare syndromes.

In the general population, germline genome-wide average mutation rates have been estimated at ($-1$–$1.5 \times 10^{-8}$ mutations per base pair per generation) from analysis of DNMs, based on whole genome sequencing (WGS) of parent-child trios and extended pedigrees[13–16]. Parental age and sex (higher burden from fathers)[13–17], and exogenous DNA damage, including exposure of a parent to ionising radiation[18] or chemotherapy[19], have been shown to elevate the number of DNMs. There is however substantial intra- and inter-family variation in the rate of accumulation of mutations. This is largely unexplained[13–17].

Colorectal cancer (CRC) is the major phenotype in several multi-tumour predisposition syndromes caused by inherited DNA repair defects that increase the rate of base substitutions and/or small indels in somatic tissues. The DNA repair processes affected are the following: (i) mismatch repair (MMR), including Lynch syndrome and constitutional MMR deficiency (principally caused by germline mutations in *MSH2*[20,21], *MLH1*[22–25], *MSH6*[26] or *PMS2*[27]); (ii) DNA polymerase proof-reading (*POLE* and *POLD1* exonuclease domain mutations)[28]; and (iii) base excision repair (*MUTYH*[29], *NTHL1*[30] and *MBD4*[31]). Constitutional MMR deficiency and the syndromes associated with *MUTYH, NTHL1* and *MBD4* are recessive conditions. Lynch syndrome is dominant, but requires random, and hence variable, 'second hits' (somatic inactivation of one allele). Families with germline *POLE* and *POLD1* mutations have dominant inheritance, but do not require somatic 'second hits'[9]. This provides a specific opportunity in *POLE* and *POLD1* carriers to analyse the contributions of both 'true' DNMs (arising in parents, one of whom is a gene carrier) and somatic (post-zygotic) mutations (arising in offspring who may be gene carriers themselves).

In this study, we used WGS of DNA from whole blood to examine directly the burdens of small-scale DNMs in nuclear families with germline *POLE* or *POLD1* mutations[28]. We determined whether the numbers of these DNMs were increased over those in control families. We also explored post-zygotic mutation burdens. Signatures of specific DNA repair defects were analysed to provide a more sensitive assessment of the effects of DNA repair deficiency, given the variability in DNMs reported in the general population. We also assessed *MUTYH* polyposis families[29] to explore the effects of mono-allelic and bi-allelic *MUTYH* mutations on DNMs. Our results have general importance for the role of DNMs in disease, and clinical importance for the risk of non-neoplastic disease in families with DNA repair deficiency syndromes.

## Results

### DNMs in control families
Using WGS to a median of 50X depth, we analysed DNMs in seven children from three nuclear *POLE* families, five from two *POLD1* families, nine from three *MUTYH* families, and 12 from three control families, along with both parents from each family (Table 1; Supplementary Table 1; Supplementary Figs. 1–3). DNMs were identified in the children and phased to determine parental origin (see Methods). No structural or chromosomal-scale DNMs were found. In the control families, the mean total DNM burden, comprising de novo simple nucleotide variants (SNVs) and small insertions or deletions (indels), was 61 (median = 60, range = 42–80). There was no significant association between DNM burden and age of the father (linear regression, t = −0.22, P = 0.83) or mother (t = −0.041, P = 0.97) at birth. This was

not unexpected based on the relatively narrow age ranges of the study subjects and the failure of some previous studies to find these associations[16,17].

### DNMs in case families
We assessed DNMs in POL families and compared them with the controls. Children whose parent had a germline mutation in *POLE* or *POLD1* had a significantly higher total DNM burden than controls (mean = 111, median = 93, range = 62–239; GLM, t = 3.18, P = 0.004; Table 1, Fig. 1a). This difference was accounted for by de novo SNVs, with no significant difference in the small number of indels present (t = 0.37, P = 0.71; Table 1). DNM burden was not associated with the child's germline POL carrier status (GLM, t = −0.39, P = 0.70). There was no significant association (P > 0.05, details not shown) in the POL families between DNM burden and the child's age or sex, or with parental age. A two-fold higher total DNM burden was found when the father carried the POL variant (t = 3.98, P = 0.003; Wilcoxon P = 0.05), although we note that only two children in our POL families had a carrier father. There was a significantly higher DNM burden in offspring of carriers of *POLE* L424V than *POLD1* S478N (mean DNM burden 140 *versus* 71, t = 2.89, P = 0.02; Table 1, Fig. 1a), and the DNM burden was not significantly different between *POLD1* and control families (t = 1.44, P = 0.17).

To identify parent-specific effects of the germline POL mutations, we phased DNMs using a 1000 base pair window around the identified variant and could assign about 30% of mutations on average to paternal or maternal origin (Table 1). The number of phased SNVs correlated strongly with total DNMs (Fig. 1b). In control families, a mean of 80% of phased DNMs came from the father, in line with previous findings (P < 0.0001, paired t test)[15,32,33]. We saw no evidence of an increased paternal contribution of phased DNMs with paternal age (t = −0.58, P = 0.57). In POL families, the parent with the germline mutation provided a greatly increased proportion of DNMs (GLM, t = 6.08, P = $1.1 \times 10^{-4}$, with no significant heterogeneity observed between *POLE* and *POLD1*) (Table 1; Fig. 1c). Specifically, there was a roughly three- to four-fold increased DNM burden from the carrier parent compared with controls, and hence a greater increase in the absolute burden from the carrier father. Of note, in our *POLD1* families, for all of whom the mother was the carrier, we saw a significant increase in the proportion of maternally derived mutations compared with control families (mean 45% versus 19%, t = −8.30, P = $1.1 \times 10^{-6}$), strongly suggesting that the DNM burden was indeed raised by the germline *POLD1* mutation. The carrier-specific DNM burden remained significantly higher in the *POLE* than *POLD1* families when the analysis was restricted to carrier mothers (mean 56 versus 30, t = 3.58, P = 0.007, Wilcoxon P = 0.03). Overall, where the mother carried the POL mutation, the proportion of maternally derived DNMs did not differ significantly from 50% (t = −0.48, P = 0.65). Thus the effect of a maternal germline *POLE* or *POLD1* mutation was to raise the DNM burden to about that of a POL-wildtype father.

There was no significant difference in the number of protein-coding region DNMs when comparing POL families and controls (t = −1.09, P = 0.29 overall; t = −0.22, P = 0.83 for *POLE*; Fig. 1d). Fewer than five coding DNMs per child were typically found and none had predicted deleterious effects. POL families had a significantly higher proportion of DNMs (mean = 17%) in late replicating (mostly non-coding) regions compared with controls (t = 2.20, P = 0.039; Fig. 1e).

### Mutational spectra and signatures
We examined six-channel single base substitution mutation spectra in the *POLE* and *POLD1* families compared with controls (Supplementary Table 2; Fig. 1f). *POLE* children had increases in the burden of all mutations, formally significant for C:G > A:T (P = 0.0004, Wilcoxon test), T:A > C:G (P = 0.0086) and T:A > G:C (P = 0.0004). However, the proportion of C:G > T:A mutations was actually reduced in the *POLE*

**Table 1 | Study participants and DNM burdensh**

| Child ID | De novo SNV burden | De novo indel burden | De novo total burden | Child genotype | Child age | Child sex | Parent with germline DNA repair variant | Age father | Age mother | Proportion phased SNVs | Proportion phased to mother | Proportion phased to father |
|---|---|---|---|---|---|---|---|---|---|---|---|---|
| POLE_A:II.1 | 93 | 4 | 82 | POLE carrier | 35 | Female | Mother | 30 | 23 | 0.37 | 0.68 | 0.32 |
| POLE_A:II.2 | 119 | 1 | 120 | POLE carrier | 43 | Female | Mother | 32 | 25 | 0.25 | 0.63 | 0.37 |
| POLE_A:II.3 | 156 | 7 | 163 | WT | 39 | Female | Mother | 37 | 30 | 0.24 | 0.37 | 0.63 |
| POLE_B:II.1 | 158 | 3 | 161 | WT | 45 | Female | Father | 22 | 19 | 0.27 | 0.05 | 0.95 |
| POLE_B:II.2 | 231 | 8 | 239 | POLE carrier | 46 | Female | Father | 24 | 21 | 0.24 | 0.07 | 0.93 |
| POLE_B:III.1 | 109 | 5 | 114 | WT | 21 | Female | Mother | 26 | 22 | 0.27 | 0.48 | 0.52 |
| POLE_B:III.2 | 84 | 4 | 88 | POLE carrier | 19 | Female | Mother | 29 | 24 | 0.27 | 0.39 | 0.61 |
| POLD1_A:II.1 | 56 | 7 | 63 | POLD1 carrier | 51 | Male | Mother | 18 | 19 | 0.32 | 0.39 | 0.61 |
| POLD1_A:II.2 | 60 | 2 | 62 | WT | 50 | Female | Mother | 20 | 22 | 0.30 | 0.50 | 0.50 |
| POLD1_A:II.3 | 65 | 6 | 71 | POLD1 carrier | 45 | Male | Mother | 24 | 25 | 0.40 | 0.46 | 0.54 |
| POLD1_B:II.1 | 76 | 7 | 83 | WT | 45 | Male | Mother | 28 | 19 | 0.36 | 0.44 | 0.56 |
| POLD1_B:II.2 | 69 | 7 | 76 | POLD1 carrier | 41 | Male | Mother | 33 | 24 | 0.17 | 0.50 | 0.50 |
| MUTYH_A:II.1 | 53 | 4 | 57 | MUTYH bi-allelic | 42 | Male | Both mono-allelic | 27 | 21 | 0.40 | 0.33 | 0.67 |
| MUTYH_A:II.2 | 60 | 3 | 63 | MUTYH mono-allelic | 36 | Female | Both mono-allelic | 30 | 24 | 0.28 | 0.06 | 0.94 |
| MUTYH_A:II.3 | 53 | 6 | 59 | WT | 35 | Male | Both mono-allelic | 32 | 26 | 0.28 | 0.07 | 0.93 |
| MUTYH_B:II.1 | 71 | 5 | 76 | MUTYH mono-allelic | 52 | Female | Mother bi-allelic | 26 | 23 | 0.25 | 0.61 | 0.39 |
| MUTYH_B:II.2 | 57 | 3 | 60 | MUTYH mono-allelic | 50 | Female | Mother bi-allelic | 29 | 26 | 0.26 | 0.73 | 0.27 |
| MUTYH_C:II.1 | 180 | 32 | 212 | MUTYH bi-allelic | 50 | Female | Both mono-allelic | 30 | 21 | 0.29 | 0.55 | 0.45 |
| MUTYH_C:II.2 | 52 | 7 | 59 | MUTYH mono-allelic | 38 | Male | Both mono-allelic | 32 | 23 | 0.19 | 0.30 | 0.70 |
| MUTYH_C:II.3 | 56 | 8 | 64 | WT | 37 | Male | Both mono-allelic | 33 | 25 | 0.36 | 0.10 | 0.90 |
| MUTYH_C:II.4 | 73 | 8 | 81 | MUTYH mono-allelic | 28 | Male | Both mono-allelic | 42 | 33 | 0.23 | 0.06 | 0.94 |
| 244:II.1 | 72 | 7 | 79 | WT | 38 | Male | | 25 | 23 | 0.25 | 0.28 | 0.72 |
| 244:II.2 | 57 | 5 | 62 | WT | 46 | Male | | 27 | 25 | 0.18 | 0.10 | 0.90 |
| 244:II.3 | 48 | 2 | 50 | WT | 47 | Male | | 29 | 27 | 0.27 | 0.08 | 0.92 |
| 244:II.4 | 42 | 5 | 47 | WT | 48 | Male | | 37 | 35 | 0.26 | 0.27 | 0.73 |
| 569:II.1 | 38 | 4 | 42 | WT | 44 | Female | | 24 | 24 | 0.24 | 0.11 | 0.89 |
| 569:II.2 | 50 | 3 | 53 | WT | 41 | Female | | 27 | 27 | 0.32 | 0.19 | 0.81 |
| 569:II.3 | 55 | 3 | 58 | WT | 37 | Female | | 31 | 31 | 0.18 | 0.30 | 0.70 |
| 569:II.4 | 66 | 9 | 75 | WT | 34 | Female | | 34 | 34 | 0.32 | 0.14 | 0.86 |
| 569:II.5 | 72 | 6 | 78 | WT | 31 | Female | | 37 | 37 | 0.35 | 0.28 | 0.72 |
| 603:II.1 | 75 | 5 | 80 | WT | 18 | Male | | 23 | 26 | 0.35 | 0.23 | 0.77 |
| 603:II.3 | 56 | 6 | 62 | WT | 28 | Female | | 31 | 34 | 0.21 | 0.25 | 0.75 |
| 603:II.4 | 49 | 2 | 51 | WT | 30 | Female | | 35 | 38 | 0.22 | 0.09 | 0.91 |

For simplicity, we refer to all nuclear families as being composed of parents and children, irrespective of current age. Full data for each child are shown, but parents are only shown in regard to their germline DNA repair mutation carrier status. All families comprise two generations except for POLE_B, which has three generations and is split into two nuclear families for our analysis (POLE_B:II.2 is the mother of POLE_B:III.1 and POLE_B:III.2). SBS burden includes a very small number of DBSs. Child age is age at blood sampling. Ages of mother and father are at the birth of the child. Proportion of phased SNVs is the proportion of all de novo SNVs that could be assigned as originating from the mother or the father (or alternatively occurring on maternal or paternal chromosomes for any post-zygotic mutations).

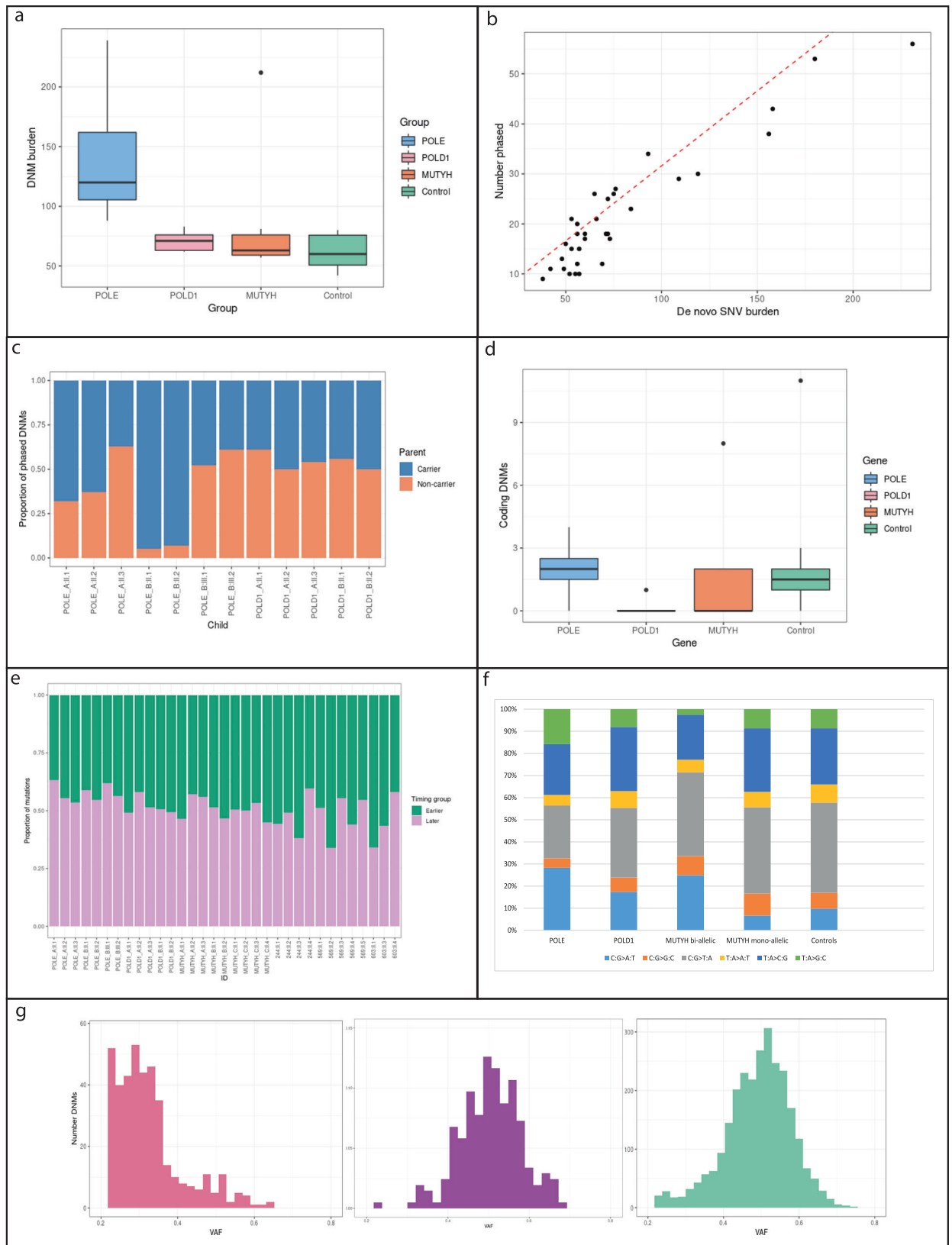

DNMs (*P* = 0.001). In *POLD1* families, only the burden of C:G > A:T DNMs was significantly increased over controls (*P* = 0.0017), with no differences in the other five channels. Since C:G > A:T mutations are the major contributors to *POLD1*-specific signatures SBS10c and 10d, these data provided further support for a modest increase in DNMs from *POLD1* carrier parents.

The commonly identified, clock-like single base substitution mutational signatures SBS1 and SBS5 were present in the DNMs from each family (Supplementary Fig. 4). When all DNMs from the *POLE* families were combined, we found two signatures characteristic of *POLE* mutations: SBS10a, enriched for TCT > TAT; and SBS28, with high levels of TTT > TGT (Fig. 2a). Interestingly, the other characteristic

**Fig. 1 | DNMs from patients and controls. a** Total DNM burdens in each of the children in POL, MUTYH and control families. Note that MUTYH families as shown here include kindreds in which one parent had bi-allelic mutations and in which both parents had mono-allelic mutations. For all box and whisker plots, boxes represent interquartile range (IQR), with the median shown as a line within the box. Whiskers are limited by values within +/−1.5 x IQR. Values outside the whiskers are shown as individual data points. **b** Association between total DNMs and phased DNMs for each child. Linear regression analysis, $y = 0.26x + 1.65$, $P = 1.17 \times 10^{-5}$, $r^2 = 0.77$. **c** Proportions of mutations phased to carrier and non-carrier parents in POLE and POLD1 families. Note that the mother is the carrier in all nuclear families except POLE:B generation II, and hence the paternal DNM excess seen in the general population is largely outweighed by the effects of the germline mutation here. **d** Number of DNMs in coding regions of genome in each of the children in POL, MUTYH and control families. Box plots are shown as for Fig. 1a. **e** Proportion of DNMs in each child mapping to early and late replicating regions of the genome. **f** Proportions of six-channel single base substitution DNMs in the five groups of study participant. **g** DNM VAF distributions for *MUTYH_C:II.1* (pink), other children from family *MUTYH_C* (purple), and all other children in the study (green). Individual distributions for all study participants are shown in Supplementary Fig. 6.

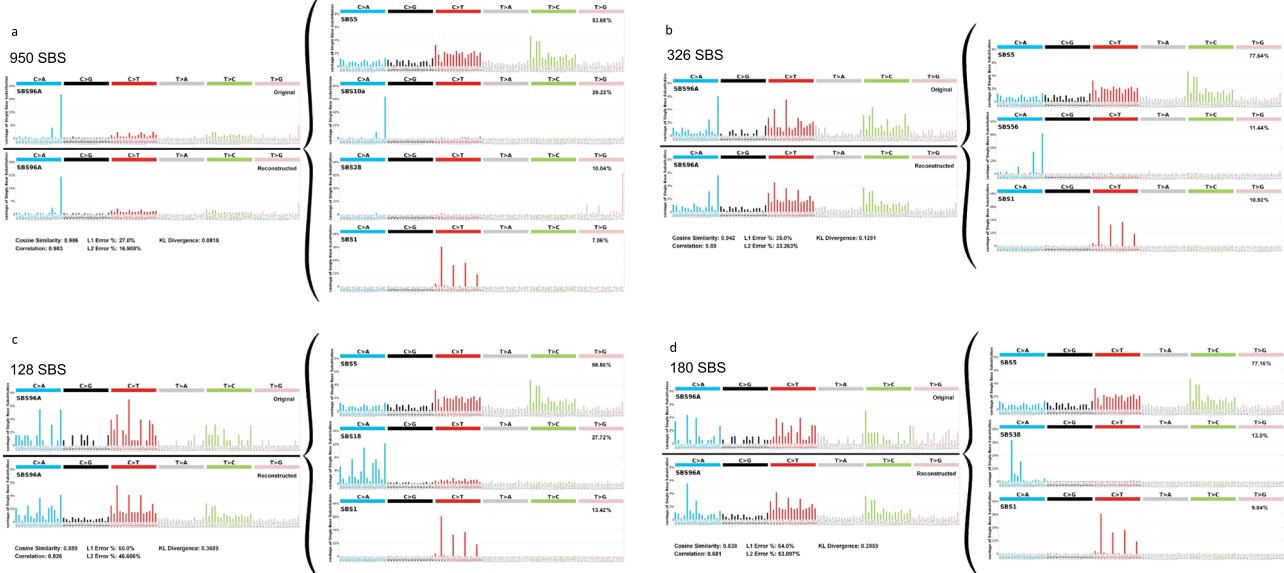

**Fig. 2 | Mutational spectra and COSMIC 96-channel mutational signatures derived from DNMs. a** All POLE families; **b** All POLD1 families; **c** family MUTYH_B; **d** Mutation signatures of all DNMs in patient *MUTYH_C:II.1*. SBS single base substitution.

*POLE* signature, 10b, typified by TCG > TTG changes, was not detected. In the combined DNMs of *POLD1* families, SBS1 and SBS5 were present, along with signature SBS56, which is formally annotated as an artefact, but closely resembles the *POLD1*-specific signature SBS10d (cosine similarity 0.98) (Fig. 2b; Supplementary Fig. 4).

We then examined mutational signatures in phased DNMs. In the combined *POLE* families, SBS10a was only found in mutations derived from carrier parents (Supplementary Fig. 5), and indeed was present only in the carrier parents in family-by-family analysis (data not shown). Whilst the number of phased DNMs in individual *POLD1* families was sub-optimal for reliable signature extraction, SBS10c was found only in the combined analysis of carrier parents (Supplementary Fig. 5).

**Post-zygotic mutations**

Whilst recognising that our methodology was primarily suited to the detection of pre-zygotic mutations, we were also able, in principle, to detect post-zygotic changes that had occurred (i) in early embryogenesis, (ii) during the development of the haematological cell system, or (iii) on a background of undetected clonal haematopoiesis (although we expected this to be rare given the ages of our subjects). We therefore explored post-zygotic mutations in POL family children. We inspected the allele frequency distributions of DNMs and found only small deviations from the symmetrical distribution around 0.5 that would be expected were all DNMs pre-zygotic (Supplementary Fig. 6). In a complementary analysis, we tested the expectation that post-zygotic mutation rates would be influenced in two ways: (1) by a hemizygous, maternally transmitted mutant POL allele in very early

embryogenesis[34]; or (2) by a heterozygous POL mutation thereafter as the offspring genome produced mRNA. Since almost all of our carrier parents were female, we were unable to distinguish between these two possibilities. We therefore performed a comprehensive exploration for an association between personal mutation carrier status and DNM burden in children, having excluded the kindred with the father carrier. Despite using several minimum DNM allele frequency cut-offs below 0.3 to simulate the sub-clonality of post-zygotic mutations, DNM burden did not differ significantly between carrier and non-carrier offspring (Wilcoxon test and logistic regression, $P > 0.05$ in all cases). The data were therefore consistent with only occasional detection of post-zygotic DNMs by our methods, implying a low burden of mutations even when the embryo expressed only the maternal mutant *POLE* or *POLD1* allele.

**DNMs in carriers of MUTYH mutations**

We next assessed DNMs from parents with mono-allelic (heterozygous) or bi-allelic (homozygous or compound heterozygous) germline *MUTYH* mutations, noting that only the latter have been reliably linked to functional base excision repair deficiency and to a tumour phenotype in humans. Our data set consisted of two families in which both parents were *MUTYH* heterozygotes and one in which the mother had bi-allelic mutations, their children comprising two bi-allelic mutation carriers, five heterozygotes and two wild-type individuals. Taking all children from the three families together, there was no excess of DNMs over controls ($t = 0.64$, $P = 0.53$), no increase in *MUTYH*-associated associated C:G > T:A mutations ($P = 0.31$, Wilcoxon test), no significant difference in the number or proportion of any

other mutation type ($P > 0.05$, $t$ and Wilcoxon tests), and no evidence of *MUTYH*- or oxidative damage-associated signatures (Table 1; Supplementary Fig. 4). There were also no differences in the numbers of DNMs in coding regions in *MUTYH* families compared to controls ($t = −0.39$, $P = 0.70$; Fig. 1c), or in the location of DNMs in relation to early or late replicating regions of the genome, after controlling for DNM burden ($t = 1.22$, $P = 0.24$; Fig. 1d).

We focussed on the *MUTYH* family (*MUTYH*_B)- in which the mother was a bi-allelic mutation carrier and her offspring were heterozygotes. Whilst there was no significant overall excess of DNMs in this family (Table 1), the children had significantly raised burdens of *MUTYH*-associated C:G > T:A mutations compared with children of *MUTYH* heterozygote parents or controls ($P < 0.05$, Wilcoxon test for both; Supplementary Table 2; Fig. 1f). Furthermore, the proportions of maternal DNMs (estimated means 45% and 67% respectively) were higher than in other *MUTYH* families (25 and 21%) and controls (12 and 19%) ($P < 0.05$, Wilcoxon test; Table 1). This result strongly suggested an impact of germline *MUTYH* deficiency on DNMs that was obscured by chance variation in DNM burden of the non-carrier parent in *MUTYH*_B. Supporting this contention, only in *MUTYH*_B did DNM mutation signatures include SBS18, a signature closely resembling the *MUTYH* deficiency signature SBS36 (cosine similarity = 0.91) (Fig. 2c)[35].

### Post-zygotic mutations in an individual treated with chemotherapy

DNM allele frequency plots suggested at most a low frequency of detectable post-zygotic mutations in *MUTYH* families, as per the POL families (Supplementary Fig. 6). To assess very early post-zygotic DNMs, we compared the four children who inherited the wildtype *MUTYH* allele from a heterozygous mother with those who inherited a mutant allele (Table 1). No evidence of a DNM excess in the latter was found (mean 62 *v* 65, $P = 0.70$). Two children with bi-allelic *MUTYH* mutations that could cause post-zygotic mutations[10] were present in our families. One of these cases (*MUTYH*_A:II.1) had an unremarkable DNM burden ($N = 57$) compared with controls, suggesting no influence of the heterozygous parents on pre-zygotic mutations, or an excess of detectable post-zygotic mutations. By contrast, the other child with bi-allelic mutations (*MUTYH*_C:II.1) had an extremely high DNM burden (180 SNVs and 32 indels), much greater than others whose parents carried mono-allelic *MUTYH* mutations, including her siblings. Ninety-five of *MUTYH*_C:II.1's mutations had an allele frequency significantly lower ($P < 0.05$) than the 50% expected for true DNMs, suggesting that some of these variants were actually post-zygotic rather than true pre-zygotic DNMs (Fig. 1g). *MUTYH*_C:II.1 had been excluded from the main study analysis, because she had received 5-fluorouracil (5-FU) for a colorectal cancer prior to blood sampling for research. However, agnostic analysis of her DNMs failed to identify 5-FU mutational signatures[36,37] (Supplementary Fig. 7). Instead, the predominant non-clock-like signature was SBS38. This signature has a hypothesised link to indirect effects of UV light. However, SBS38 often co-occurs with oxidative damage signature SBS18[38] and is characterised by C > A mutations that can result from the 8-oxo-G:A mispairs that are classically corrected by MUTYH. We hypothesised that the chemotherapy had selected for oligoclonal haematopoiesis, which in turn revealed somatic mutations resulting from *MUTYH* deficiency[39–45]. Further investigation supported the existence of such a leukocyte clone, admixed with a smaller clone of true DNMs (Supplementary Fig. 7). We estimated that *MUTYH*_C:II:1 had no excess of true DNMs ($N = 64$), and only clock-like signatures were present in this set of mutations, without evidence of oxidative damage signatures. The bulk of mutations was in the putative leukocyte clone, and this additionally showed signatures SBS9 (polη in leukocytes), SBS36 and SBS38, all consistent with oligoclonal haematopoiesis on a background of MUTYH deficiency (Fig. 2d). We found no driver mutations in *TET2, DNMT3A, PPM1D, IDH1,* *IDH2, TP53, CHEK2, ASXL1, MBD4* or other genes associated with myelodysplasia or AML, and the patient has reported no haematological problems in over two decades of follow-up. The clonal haematopoiesis was thus not necessarily neoplastic and perhaps resembled that found in normal aging[46].

## Discussion

The issue of whether individuals with DNA repair defects create an excess of DNMs in their children is a longstanding one, but has not previously been addressed directly. For example, it has not been clear whether the occasional reports of developmental problems or other non-cancer diseases in the children of such families are the result of DNMs, or other factors including selection bias and chance. We now show that DNMs are most unlikely to be the cause where the defect involves failure to correct mispaired or modified bases. Parental germline defects in DNA polymerase proofreading do lead to an excess of (pre-zygotic) DNMs, especially base substitutions, in offspring, with a stronger effect in carriers of *POLE* L424V than carriers of *POLD1* S478N. The effect occurs in parents of both sexes, but is greater in terms of excess mutation numbers in fathers. The effects are, however, quite modest, with roughly a doubling or trebling of DNM burden where parents carry *POLE, POLD1* or bi-allelic *MUTYH* mutations. Thus, in mothers, the effect of the germline mutations is only to raise the DNM burden to the typical level of a wild-type father (Table 1). Furthermore, <5% of the DNMs occur in coding regions, which is consistent with the lack of deleterious phenotypes in our families (apart from the tumours that arise owing to the inherited defects in DNA repair). It is possible that some mutations called as DNMs actually occurred in the very early zygote, and it is likely that children carrying polymerase proofreading defects continue to accumulate mutations in somatic tissues[9]. We expected to detect an excess of post-zygotic (somatic) mutations in child POL gene carriers—for example, arising in very early embryogenesis or haematopoietic stem cells—but such an effect was small and/or below the resolution of our assays.

On average across all families, 28% of DNMs could be phased. Whilst we showed that the proportions of DNM assigned to each parent were accurate (see Methods), the numbers of mutations phased to individual parents were sometimes sub-optimal for signature extraction. This was especially the case in controls and in non-carrier parents, where fewer than 20 mutations were typically phased successfully (Table 1). The small number of phased DNMs caused low stability in the signature decomposition output for individual nuclear families, and we were mindful to interpret signature outputs with caution. Despite this, pooled analysis specifically identified POL-associated signatures in the carrier parents, with the surprising exception of SBS 10b, which is one of the three characteristic *POLE* SBS signatures and is typified by CpG>TpG changes (Fig. 1a). The reasons for the absence of SBS 10b are unclear, but since this signature was derived from tumours and is present in the normal colorectum of *POLE* cases[9], its absence from DNMs may well reflect true differences in mutational processes or DNA repair between the germ line and soma[17].

The *MUTYH*-associated polyposis syndrome has recessive inheritance. We found no evidence of an increased DNM burden in children of *MUTYH* heterozygotes, consistent with the very modest or zero increased risk of cancer reported for those individuals. Whilst we failed to find a significant excess of DNMs in the two offspring of the parent in our study with bi-allelic *MUTYH* mutations (family *MUYTH*_B), phasing of mutations and signature analysis indicated that there was an excess DNM burden from the carrier mother, of a magnitude similar to that in POL families. The failure to detect this by bulk DNM analysis reflected the small number of individuals, the lower DNM burden arising from mothers, and the natural variability in DNM burden in controls. We conclude that parental *MUTYH* deficiency probably does elevate the number of DNMs carried by offspring, but, like *POLE* and *POLD1*, the effects appear to be modest.

The case of *MUTYH*_C:II.1 provides a previously unreported insight into how chemotherapy and DNA repair deficiency can interact to produce an expanded and detectable leukocyte clone that does not carry a burden of mutation caused by the 5-FU therapy given over 20 years previously. A corresponding lack of a 5-FU signature in therapy-induced AML was noted by the elegant study of Pich et al.[45], who suggested that the AML precursor cells are quiescent and hence unaffected by 5-FU. *MUTYH*_C:II.1's blood carries an unusual mutational spectrum, dominated by C > A mutations, only some of which are in a typical trinucleotide context for *MUTYH* deficiency. This signature appears consistent with unrepaired oxidative damage, and we speculate that it resulted in part from an atypical set of original mutations caused by a 5-FU-induced nucleotide pool imbalance. In platinum-treated patients, Pich et al.[45] found that clonal haematopoiesis (without overt AML) was likely to pre-exist and to be revealed by chemotherapy rather than caused by it. Pre-existing clonal haematopoiesis is arguably unlikely in *MUTYH*_C:II.1 given the patient's age at treatment (26 years), and we found no evidence of somatic driver mutations in *MUTYH*_C:II.1's DNA. We therefore propose that in *MUTYH*_C:II.1 the microenvironment—for example, a lack of competing clones owing to extinction of most lineages by 5-FU—reduced hae-matopoietic clonal competition. This allowed re-population of the stem cell pool by a small number of surviving clones as leukocyte numbers re-expanded, through drift or some unknown selective advantage. The mutations resulting from *MUTYH* deficiency were thus present in a large proportion of leukocytes and detectable more than two decades later at blood sampling.

DNM calling is improving, but remains challenging. Given the small numbers of DNMs identified per individual, even small imperfections in sequencing quality, mapping or calling can produce a major effect genome-wide. In order to mitigate these problems, we generated two independent libraries for each of our study participants' DNAs, and we sequenced all samples contemporaneously using the same sequencing and analysis pipelines. Nevertheless, DNM validation still required laborious visual inspection to ensure quality (Supplementary Fig. 2). The resulting estimates of DNM burdens and parent-specific effects in controls were in line with previous estimates[15,32]. With the *caveat* of different platforms and analysis pipelines, the DNM burdens in our DNA repair-deficient families also overlapped with those of 12 individuals with raised DNM burdens from a total of over 20,000 cases with developmental disorders and other non-cancer phenotypes reported by Kaplanis et al.[19]. Bi-allelic germline DNA repair defects were subsequently identified in the fathers of two of these 12 individuals. One father carried a homozygous loss-of-function mutation in a known disease gene, *XPC*, and the DNM mutation spectrum closely resembled that expected, but selective or mutational effects of parental chemotherapy on the DNM burden could not be excluded. Furthermore, in neither case could the DNMs detected be linked to the phenotypes of the children, which were specifically intellectual disability and epileptic encephalopathy. Whilst recognising that our work might have benefitted from a larger sample, especially for the *MUTYH* families, it was large enough to draw general conclusions about the magnitude and importance of increased DNM burdens in families with inherited DNA repair defects as the base pair level.

In summary, we have shown that inherited defects in the repair of base substitutions and small insertion-deletion mutations not only predispose to cancer through raised somatic mutation rates, but also increase the burden of de novo germline mutations. Gamete production and very early embryogenesis, in which maternal genes are exclusively expressed for the first four cell divisions, are clearly dependent on DNA replication and cell division, and the excess of DNMs in the offspring of POL mutation carriers and the presence of POL-specific signatures can thus be explained through replication errors. Our data suggest that oxidative damage, which is classically repaired by *MUTYH*, also occurs in the germ line, despite what might be

thought to be a low exposure environment[47]. Syndromes caused by hypermutation and/or failure to repair DNA damage are hetero-geneous, and as a result, the DNM risk may vary. Nevertheless, parents with defective DNA repair of any type frequently express worries that their children will inherit a predisposition to other diseases, as well as the increased cancer risk. Whilst the latter is hard to avoid, it is reas-suring for families with DNA repair problems at the base pair level that the increased DNM burdens are modest, DNMs are rarely found in coding regions of the genome, and phenotypic consequences are predicted to be rare.

## Methods

Six families with germline *POLE*, *POLD1* or *MUTYH* mutations, including one with three generations affected, were recruited by the CORGI study[48] and one by the University Hospital, Basel (Supplementary Fig. 1). No child had received genotoxic therapy prior to blood sampling, with the exception of patient *MUTYH*_C:II.1 who was removed from the main study and analysed separately. No parent had received genotoxic therapy prior to having children. Three control families were from the Generations Scotland/Scottish Family Health Study[49] and had no known germline mutations in DNA repair pathways or other inherited disorders. When describing our results from our eight nuclear families, we refer to "parents" and "children/offspring" irre-spective of their age. All children, bar one subsequently excluded from the main study, were chemo/radiotherapy-naive at the time of blood sampling and no parent had received genotoxic treatment prior to birth of their children. There were no other notable non cancer/poly-posis phenotypes. CORGI and CORGI2 were approved by South Cen-tral Hampshire A Research Ethics Committee and South Central Oxford A Research Committee, references 17/SC/0079, 06/Q1702/92 respectively. Ethical approval for the GS:SFHS study was obtained from the Tayside Committee on Medical Research Ethics (on behalf of the National Health Service), reference 05/S1401/89. All patients provided written, informed consent to taking part. Participation in this research raised no issues related to Inclusion and Ethics.

Peripheral blood was sampled from each participant and con-stitutional DNA extracted. Two independent PCR-free libraries were constructed per person and each was sequenced to a mean depth of 25X (median 25X, range 22–31X). The raw fastq files were aligned to the hg19 reference genome using BWA (v 0.7.16)[50] and adapter sequenced trimmed using CutAdapt (v 1.9.1)[51]. Duplicate reads were marked using Picard (v2.17.11)}[52] and the BAMs from independent library preparations from the same DNA samples were combined using Picard (v 2.17.11) to make BAM files with ~60X coverage per individual (median 49X, range 45–56X). We used the Genome Ana-lysis Toolkit (GATK) (v 4.0.10.1)[53] for further BAM file processing and germline variant calling from autosomes only, following the recom-mended best practices pipeline[54]. Following construction of per individual gVCFs, variants observed were joint genotyped across individuals from the same family. Per site depth calculations were performed using Samtools (v1.6)[55]. Sites in the top 0.01% of read coverage were excluded from further analysis as thousands of reads aligned to these sites.

De novo mutations in parent-offspring trios were called using DeNovoGear (v 1.1.1)[56], which jointly analyses the likelihood of the genotypes at any genomic site in a trio (using default mutation rate priors and posterior probability threshold). The resulting putative DNMs were filtered using quality control steps to remove false positive DNMs (Supplementary Fig. 2). Since previous work had demonstrated that the vast majority of real germline DNMs have alternative allele read support between 30 and 70% variant allele frequency (VAF)[33] and <1% read support in either parent, DNMs that were retained for analysis had a minimum read depth of 20 in all members of the trio, were called by GATK in the child but not in either parent, had a VAF of 30–70% in the child and <1% variant read support from either parent. DNMs

mapping to simple repeats, segmental duplications and the human leukocyte antigen (HLA) region were also excluded. Ensembl Variant Effect Predictor (VEP) (v 97)[57] was used to annotate DNMs and only those with a frequency below 1% in the gnomAD non-Finnish European population dataset were retained. Finally, all putative de novo variants passing these filters were assessed in the Integrative Genomics Viewer (IGV) browser (v 2.3.900)[58], resulting in the exclusion of a further ~20% of calls, largely owing to poor local sequence quality likely to have resulted from mapping errors. We additionally utilised the somatic mutation calling mode of Mutect2 within GATK to assess mutations of putative post-zygotic origin.

DNMs were phased to identify the parental gamete of origin using DeNovoGear. A window size of 1000 base pairs around the de novo variant was set to search for nearby SNVs present in one parent and therefore informative of read inheritance. Increasing the window size to 3000 base pairs to look for phase informative SNVs did not increase the number of mutations that could be phased. Phasing using https://github.com/queenjobo/PhaseMyDeNovo as used by Kaplanis et al.[19] also did not increase the number of mutations that could be phased. Since family POLE_B had three generations, we could directly phase DNMs observed in POLE_B:II to the generation I parent of origin using generation III haplotypes in order to check DeNovoGear phasing accuracy. We used the duoHMM phasing method in SHAPEIT v2.r837[59] with 1000 Genomes phase I reference panel to predict haplotypes whilst taking into account the family structure. Of 239 DNMs called in POLE_B:II.2, 165 (69%) were inherited by at least one of her children (POLE_B:III.1 and POLE_B:III.2). We found that all 38 DNMs phased by DeNovoGear were assigned to the correct parent. Furthermore, for the 127 DNMs assigned by SHAPEIT alone, the proportions assigned to each parent were very similar to the mutations assigned by DeNovoGear (92% paternal, 8% maternal with DeNovoGear; 91% paternal, 9% maternal with SHAPEIT).

Python package SigProfiler (v 1.1.3) (13) was used to decompose the SBS mutational signatures of the DNMs and fit them to the COSMIC v3.2 reference signature set. Cosine similarities between signatures were calculated in R (v 4.1.0). Signatures were analysed for (i) groups of patients by genotype or case-control status, (ii) all DNMs by individual, and (iii) phased DNM by parent sex and carrier status.

DNA replication timing data for the CRC cell line HCT116 had been generated previously (https://doi.org/10.7488/era/2637). Quantitative replication timing estimates for 10,000 bp windows of the genome were obtained following a modified Repli-Seq method described by Marchal et al.[60]. Briefly, two replicates of HCT116 cells were treated with the thymidine analogue EdU, fixed in ethanol and stained with propidium iodide—allowing cells to be separated according to cell cycle stage by fluorescence-activated cell sorting (FACS) into in early, mid and late S-phase fractions (45). DNA with EdU incorporated was then immunoprecipitated using Click chemistry and libraries were generated before being sequenced. Repli-seq data were analysed as described and the replication timing (T) value for each 10,000 bp region was defined as the ratio of reads per million in early S-phase compared to late S-phase

$$T = \ln(\text{Early}/\text{Late})$$

Replication timing values across the genome were then smoothed using quantile normalisation, in order to reduce noise.

Association tests were performed using $t$ tests, supplemented by Wilcoxon tests where few observations were available, or linear regression, as indicated in the manuscript. Multivariable analyses were performed using multivariable logistic or generalised linear model (GLM) regression. Since the size of each family was too small to take into account potential family-specific variation in DNMs, each parent–parent–child trio was treated as if an independent data point.

## Reporting summary

Further information on research design is available in the Nature Portfolio Reporting Summary linked to this article.

## Data availability

Control genomic data can be requested from the Scottish Family Health Study (Generations Scotland) by email to access@generationscotland.org. Details of the application procedure can be found at https://www.ed.ac.uk/generation-scotland/for-researchers/access. Patient data can be requested from ian.tomlinson@oncology.ox.ac.uk, with response expected within four weeks. All data are protected by formal agreements in order to preserve patient anonymity and privacy, and to comply with ethical permissions. Data will be released to researchers subject to formal compliance with these conditions of anonymity and existing ethical permissions, as incorporated into a data transfer agreement based on the standard models used by the host institutions concerned.

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

## Acknowledgements

We thank all of the clinicians and healthcare providers who have assisted in recruiting patients to CORGI and CORGI 2.0 and obtaining samples and clinical information. We would also like to thank all of the patients and their families who consented to the use of their samples and medical information for the purposes of this study. We also acknowledge several helpful comments made those who reviewed the manuscript for the journal. Generation Scotland received core support from the Chief Scientist Office of the Scottish Government Health Directorates [CZD/16/6] and the Scottish Funding Council [HR03006], and we are grateful to the general practitioners and the Scottish School of Primary Care for their help in recruiting them, and the whole Generation Scotland team, which includes interviewers, computer and laboratory technicians, clerical workers, research scientists, volunteers, managers, receptionists, healthcare assistants and nurses. I.T. acknowledges Cancer Research UK Programme Grant (C6199) and ERC EVOCAN project funding. C.P. acknowledges Bowel Cancer UK funding (grant 18PG0010). D.S. is a Cancer Research UK Career Development fellow (reference C47648/A20837), and work in his laboratory is also supported by an MRC university grant to the MRC Human Genetics Unit. I.K. was funded by a studentship from Cancer Research UK as well as the A.G. Leventis Foundation.

## Author contributions

I.T. and C.P. designed the study. L.M., J.R.S., A.D., A.C., A.G., K.-R.O., K.H., M.N., H.T., A.L., C.P. and I.T. recruited patients and obtained clinicopathological data. K.S. collated and analysed data, with assistance from C.P. and I.T. J.C.W., I.S., I.K. and D.S. contributed to the analysis of the replication timing data. R.R. provided advice on DNM analysis. K.S., C.P. and I.T. wrote the manuscript, taking account of comments from all other authors.

## Competing interests

The authors declare no competing interests.
