## [Peer Review File · Nature Communications]

Germline de novo mutations in families with Mendelian cancer syndromes caused by defects in DNA repairREVIEWER COMMENTS

Reviewer #1 (Remarks to the Author):

In their report, Sherwood et.al. investigate the effects of de novo mutations in the germline of individuals born to parents with defects in DNA repair genes. The study utilizes a cohort of participants wherein the parents are either carriers of replicative polymerase POLE and POLD mutations or are mono/biallelic for MUTYH mutations which plays a central role in mismatch repair.

Compared to control families, individuals from POLE/POLD families had a higher mutation burden, although there was no significant correlation between mutation burdens and the age/sex of children, germline POL status of children, parent age or father's age. Children of POLE carriers additionally showed a replication timing bias for the DNMs, suggesting that POLE dysfunction during earliest divisions might play a minor role in generating some germline mutations. Despite the low mutation burden in the germlines, the authors were nevertheless able to derive some meaningful signatures from pooled datasets, including the ubiquitous SBS1 and SBS5 signatures and replicative-defect associated SBS10 signatures in the POLE dataset.

However, there weren't sufficient mutations for accurately determining INDEL or DBS signatures for most of the datasets. Finally, the MUTYH deficiencies were largely neutral vis-à-vis mutation burden, save for a single dataset which correlated with prior chemotherapy and clonal hematopoiesis.

In this reviewer's opinion, this is an important study that contributes to our understanding of genomic human health predictors. While somatic hypermutations are widely studied, de novo germline mutations and their contribution to disease phenotypes are often grossly overlooked. The data shows an overall low germline DNM burden and modest-to-no phenotypic consequences in response to germline repair defects, in either parents or the individuals. This suggests that genome maintenance within germlines is likely governed by robust surveillance. Such a mechanism likely counter-selects against deleterious mutations and chromosomal aberrations.

Overall, the manuscript is well-written, although the mutational analysis could do with a bit more detail, as noted below in the comments:

- 1) My main concern is that the authors used an older version of the human genome for their alignments. I recommend converting to hg38 for more accurate SNV calling for its better genomic resolution. I suspect the authors would find a fair bit of genotype discordance between the two versions, and in the process, might end up finding many more mutations, which could be useful for downstream signature analysis etc.
- 2) While signature analysis is nice, such data often has substantial background noise which hinders interpretation. It would be useful to see the actual mutational spectrum of the DNMs in the study sets, divided by C→T changes etc. A graph would be quite nice. In addition, please provide a supplementary table containing the variant allele information.
- 3) As a follow up to the identified CpG→ TpG changes seen with signature 10b as well as SBS1, it would be useful to check the position of DNMs across the genome to see if they correlate with DNA methylation status. Methylated sites have a high degree of spontaneous deamination, which could result in C→T changes at CpG sites. ENCODE should have some useful datasets for comparisons (e.g GM12878). POLE mutator effect at methylated CpGs has been recently described (Poulos et.al 2017, PMID: 28531315)
- 4) In Table 1, please indicate p-values for the differences in proportions of phased mutations from the mother v the father

5) A useful addition to the POLE dataset analysis would be the ascertainment of replication strand bias. There are several excellent ways to accomplish this including recent work from Julian Sale's group (Murat et.al 2022, PMID: 36351018).

6) Please include a citation for the POLE and POLD mutants at first instance (Results line 136, should be a straightforward self-citation)

7) Introduction para 5 "Colorectal cancer (CRC) is the major phenotype in several..."- please add appropriate references for all the DNA repair processes involved.

8) Supplementary Fig-4 - Define the statistical test for significance analysis in the legend.

9) The acronym "GLM" shows up in multiple references of statistical analysis without definition. Please define at first usage.

10) Typo in Lines 274-275- "... less than 5% of mutations are in the non-coding regions". Should say coding regions.

11) I didn't notice a statement of data sharing/availability for the sequenced datasets within the main manuscript. If available, please provide the relevant accession numbers or put in placeholders for when these become available.

Reviewer #2 (Remarks to the Author):

In this manuscript, Sherwood et al investigated the impact of defective DNA repair genes in carrier families on the de novo mutation landscape of their offspring. The study focused on seven families with germline mutations in one of three genes, namely POLE, POLD1 and MUTYH. A recent study by Kaplanis et al (Nature 2022), examined a related question using a much larger unselected cohort of trios, but this current study is unique in the specific recruitment of these DNA repair deficient families for analysis. The limited number of samples and de novo mutations preclude further detailed investigation of mechanisms and strong conclusions to be made, nevertheless the data does support the main conclusion that few de novo disease causing variants should arise in these individuals.

Overall, the findings are somewhat descriptive, but given the rarity of such cohorts, the findings will still be informative for the field. The basic analysis of the data (i.e. mutation calling) appears technically sound, however there are a number of issues with the interpretation and presentation of some of the data.

(1) The abstract states that there are 8 families, but only 7 has been analysed and are presented in the paper.

(2) In the abstract, the statement "DNM burdens arising from parents of either sex who carried a single germline POLE or POLD1 mutation, or biallelic MUTYH mutations, were approximately 3- to 4-fold increased over controls." is not correct. Based on the VAF distribution in Sup Fig 8, the mutations of MUTYH_C:II.1 (the biallelic MUTYH offspring) are likely post-zygotic and probably arose during clonal haematopoiesis. Therefore, they should not be considered de novo mutations. As such, based on the presented data, only offsprings with a POLE carrier parent have increased DNM (and more like 2-3 fold rather than 3-4 fold).

(3) The bulk of the main results is summarised in Table 1. While this table is useful and informative, I feel that the description of the results would be easier to follow if some of the findings and comparisons could be presented in simple bargraphs with error bars and significance where relevant.

(4) Line 142 "The number of phased SNVs correlated strongly with total DNMs (details not shown)."

Related to the above comment, this can easily be shown as a scatterplot, perhaps as a supplementary figure.

(5) SBS38 is not particularly strong in MUTYH_CII.1. How many mutations are attributed to this signature? De novo signature extract should be performed to determine if this signature is readily detectable.

(6) Similarly, for the bottom panel in Fig2, I don't think matching to ID signatures would be very reliable. For instance, although MUTYH_AII.1 has ~0.6 similarity to ID6, the sample only has 4 indels.

(7) Generally, the presentation of the mutational signatures is misleading as the signatures themselves are just taken from COSMIC with a % contribution assigned by SigProfiler. For the mutational signature analysis, the original trinucleotide mutational spectrum, along with the residue error after COSMIC signature fitting, should be provided. De novo extracted signatures should be shown.

(8) It would be useful to include a supplementary table showing the chemotherapy that each patient may have been exposed to.

(9) Panel labels are missing in Fig 2 and the figure legend of (c) doesn't seem to match what is shown in the figure.

(10) The results section could benefit from some subheadings to improve clarity.

(11) Introduction line 91. I think a more commonly used term for biallelic MMR deficiency is constitutional MMR deficiency rather than congenital MMR

(12) Line 96 "Families with germline POLE and POLD1 mutations provide a specific opportunity..." Provide a specific opportunity for what?

(13) Line 379, 382 Missing references

Jason Wong

Reviewer #3 (Remarks to the Author):

While germline mutations in DNA repair genes are known to contribute to increased mutagenesis in tumors, it remains an open question to what extent they contribute to de novo mutations in normal tissues from carriers and their offspring. In the current study, Sherwood and colleagues address this question by analyzing blood WGS from carriers of germline POLE, POLD1, and MUTYH mutations parents and their offspring from several independent families. They make several notable findings including that the overall increase in pre-zygotic DNMs is roughly 2- to 4-fold elevated over control families, but that this increase is almost entirely in non-coding regions of the genome. They also find that while DNMs are not elevated overall in POLD1 offspring, when the DNMs are phased to the parent there are detectable increases and this is affected by the sex of the carrier parent. They also make an interesting finding in post-zygotic DNMs in a MUTYH carrier who underwent 5FU treatment. Robinson et al (Nature Genetics 2021) previously sequenced a number of normal tissues including intestinal crypts, endometrial glands, skin, sperm, and blood from fourteen POLE/POLD1 carriers. While the authors do cite this study on several occasions, this manuscript is written in such a way that a reader could be forgiven for not realizing that such POLE/D carrier normal tissue sequencing had been done. It strikes this reviewer as a curious missed opportunity to address some of the strengths of this study.

Issues to be addressed:

The signature analysis was performed on pooled mutation data. The TCT mutations account for roughly ~2/3 (I apologize but the font is incredibly small on the y-axis label) of the 30% of total mutations attributed to SBS10a, meaning approximately ~20% of the total mutations in the pool are that POLE signature. Likewise SBS28 is 45% of 10% (4.5% total mutations). So the post-zygotic POLE-specific DNMs represent ~25% of total mutations when the total DNMs are increased 3-4-fold. An increase from 60 to 120 is a 60 DNM increase per individual meaning ~30, or half of the increase, could be attributed to POLE. Could the authors speculate on what is driving the increase in non-POLE DNMs in the POLE samples?

Supp Fig 3 should really indicate each data point. Supp Fig 3a also feels a bit misleading. The data demonstrate a clear difference between POLE and POLD1. The authors do show a difference in the phased mutations and discuss that the lack of difference may result from sex and size sample biases. But the figure as presented implies differences in POLS relative to controls, when POLD1 is no different than controls.

I realize that in the legend, the author's state that the reasons are unclear, but this reviewer would be interested in hearing what is the authors' explanation for the SBS10d signature non-carrier parents (Supp Fig 6). They mention the low number of mutations and the low percentage (13%). Related to this is that SBS28 is at lower percentages in the POLE samples (10-11%). The total numbers of these mutations are not given but would perhaps be informative here.

The authors find increased DNMs only in phased mutations for POLD1 and MUTYH. But only ~25% of DNMs were able to be phased. Do the authors think this is a technical limitation and that something (increased depth, etc) would identify more POLD1 or MUTYH signatures? Or that there are simply very few pre-zygotic POLD1/MUTYH mutations to be identified? Or some other scenario?

Line 285 Missing period (".") between "extraction" and "This"

REVIEWER COMMENTS

We are grateful to all reviewers. We have performed extra data analysis in response to their comments and detail this below. The major changes comprise display of full mutation spectra and the derived signatures for each class of patient, and more figures in the main text to aid data interpretation.

Reviewer #1 (Remarks to the Author):

In their report, Sherwood et.al. investigate the effects of de novo mutations in the germline of individuals born to parents with defects in DNA repair genes. The study utilizes a cohort of participants wherein the parents are either carriers of replicative polymerase POLE and POLD mutations or are mono/biallelic for MUTYH mutations which plays a central role in mismatch repair.

Compared to control families, individuals from POLE/POLD families had a higher mutation burden, although there was no significant correlation between mutation burdens and the age/sex of children, germline POL status of children, parent age or father's age. Children of POLE carriers additionally showed a replication timing bias for the DNMs, suggesting that POLE dysfunction during earliest divisions might play a minor role in generating some germline mutations. Despite the low mutation burden in the germlines, the authors were nevertheless able to derive some meaningful signatures from pooled datasets, including the ubiquitous SBS1 and SBS5 signatures and replicative-defect associated SBS10 signatures in the POLE dataset.

However, there weren't sufficient mutations for accurately determining INDEL or DBS signatures for most of the datasets. Finally, the MUTYH deficiencies were largely neutral vis-à-vis mutation burden, save for a single dataset which correlated with prior chemotherapy and clonal hematopoiesis.

In this reviewer's opinion, this is an important study that contributes to our understanding of genomic human health predictors. While somatic hypermutations are widely studied, de novo germline mutations and their contribution to disease phenotypes are often grossly overlooked. The data shows an overall low germline DNM burden and modest-to-no phenotypic consequences in response to germline repair defects, in either parents or the individuals. This suggests that genome maintenance within germlines is likely governed by robust surveillance. Such a mechanism likely counter-selects against deleterious mutations and chromosomal aberrations.

Overall, the manuscript is well-written, although the mutational analysis could do with a bit more detail, as noted below in the comments:

1) My main concern is that the authors used an older version of the human genome for their alignments. I recommend converting to hg38 for more accurate SNV calling for its better genomic resolution. I suspect the authors would find a fair bit of genotype discordance between the two versions, and in the process, might end up finding many more mutations, which could be useful for downstream signature analysis etc.

Thanks for this useful comment. We had used B37 in large part to achieve comparability with the Hurles group's manuscripts on DNMs (Rahbari, 2016; Kaplanis, 2022), especially as we had re-analysed some of the same control samples as them and wished to assess comparability of the studies. A further factor in our decision was that GATK was optimised for B37. We accept that slightly worse performance (~2-5%) compared with B38 was to be expected (Schneider 2017;

Lowy Gallego, 2019), but felt this to be an acceptable trade-off, since all participants were sequenced together using the same pipeline. We have now re-mapped and called three individuals from family POLE_A using B38, and compared results with B37. The results are shown below for SBS DNMs (B37 blue, B38 red).

On average, the number of DNMs identified by B38 is similar to B37 (within 10% either way), with 80% overlap of specific DNMs. There is evidently no clear DNM increase in the B38 analysis in this small sample. Visual inspection to check and assess the DNMs suggested that the differences between builds resulted from quality score differences, presumably reflecting the interplay between our sequencing errors and errors in the reference sequences.

We therefore request that we retain the B37 analysis. We have added a supplementary figure showing the B37 v B38 data above.

2) While signature analysis is nice, such data often has substantial background noise which hinders interpretation. It would be useful to see the actual mutational spectrum of the DNMs in the study sets, divided by C→T changes etc. A graph would be quite nice. In addition, please provide a supplementary table containing the variant allele information.

We have added back in the raw mutation spectra and a Supplementary Table 2 of six-channel mutations for POLE, POLD1 and controls, with channel proportions shown graphically in Figure 1f.

The variant lists will be made available as part of the data release as per journal policy.

3) As a follow up to the identified CpG→TpG changes seen with signature 10b as well as SBS1, it would be useful to check the position of DNMs across the genome to see if they correlate with DNA methylation status. Methylated sites have a high degree of spontaneous deamination, which

could result in C→T changes at CpG sites. ENCODE should have some useful datasets for comparisons (e.g GM12878). POLE mutator effect at methylated CpGs has been recently described (Poulos et.al 2017, PMID: 28531315)

We have assessed the feasibility of this analysis based on several reference data sets, including GM12878, but we typically lose >90% mutations during the process of filtering DNMs. Thus, even for the most common DNMs, namely those from all POLE families, we end up with fewer than 5 mutations from each patient type that are mapped to a CpG with available methylation data. The analysis is thus not feasible in this context.

4) In Table 1, please indicate p-values for the differences in proportions of phased mutations from the mother v the father

We would argue that this adds will not add much for the case families, as it depends so strongly on who carries the DNA repair mutation(s), as we state. We have assessed these proportions in controls and found a much higher paternal proportion, as expected; we now add the formal statistical test results to the text (para. 3, page 5). We have also added to the Figure 1 legend that whilst we expect more paternal than maternal mutations, it must be noted that the mother is the carrier for all bar two POL family children.

5) A useful addition to the POLE dataset analysis would be the ascertainment of replication strand bias. There are several excellent ways to accomplish this including recent work from Julian Sale's group (Murat et.al 2022, PMID: 36351018).

Given the unresolved issues of whether wildtype or mutant polymerases can proofread both strands, coupled with their roles in DNA repair outside the context of replication, we did not perform this analysis. However, we agree that it represents is a useful check and are grateful for the suggestion. We were unable to use the available lists of the positions of specific replication origins, as few DNMs in our families mapped close enough to one of these origins, many of which are, in any event cell type-specific). We therefore used the Haradhvalah (Cell, 2016) method, which had shown strand asymmetry in large data sets. Such reference data allows almost all DNMs to be mapped, but the replication timing smoothing used probably dampens the effect size (e.g. there was only a 2-fold strand bias in POLE-mutant cancers in the original study).

	Leading	Lagging
POLE phased carrier	38	31
POLE phased non-carrier	7	18
POLD1 phased carrier	7	10
POLD1 phased non-carrier	11	13
Controls all phased	43	48

The data are consistent with the expected effects, but the differences are not formally significant.

6) Please include a citation for the POLE and POLD mutants at first instance (Results line 136, should be a straightforward self-citation)

This is now covered by the reply to 7)

7) Introduction para 5 “Colorectal cancer (CRC) is the major phenotype in several...”- please add appropriate references for all the DNA repair processes involved.

We have added these.

8) Supplementary Fig-4 - Define the statistical test for significance analysis in the legend.

The test was performed on phased v all DNMs and used GLM (now spelt out in the Methods as per comment (9)). Results are now in the main text,

9) The acronym "GLM" shows up in multiple references of statistical analysis without definition. Please define at first usage.

Now added to the Methods.

10) Typo in Lines 274-275- "... less than 5% of mutations are in the non-coding regions". Should say coding regions.

Sorry, this has been corrected

11) I didn't notice a statement of data sharing/availability for the sequenced datasets within the main manuscript. If available, please provide the relevant accession numbers or put in placeholders for when these become available.

This will be added according to journal policy, although agreements will be required to protect patient confidentiality..

Reviewer #2 (Remarks to the Author):

In this manuscript, Sherwood et al investigated the impact of defective DNA repair genes in carrier families on the de novo mutation landscape of their offspring. The study focused on seven families with germline mutations in one of three genes, namely POLE, POLD1 and MUTYH. A recent study by Kaplanis et al (Nature 2022), examined a related question using a much larger unselected cohort of trios, but this current study is unique in the specific recruitment of these DNA repair deficient families for analysis. The limited number of samples and de novo mutations preclude further detailed investigation of mechanisms and strong conclusions to be made, nevertheless the data does support the main conclusion that few de novo disease causing variants should arise in these individuals.

Overall, the findings are somewhat descriptive, but given the rarity of such cohorts, the findings will still be informative for the field. The basic analysis of the data (i.e. mutation calling) appears technically sound, however there are a number of issues with the interpretation and presentation of some of the data.

The study aimed to answer the question of whether these families had raised DNMs and, if so, to estimate the size of this effect. The description of the mutation burdens is part of that answer.

(1) The abstract states that there are 8 families, but only 7 has been analysed and are presented in the paper.

We now use the term nuclear family in the Abstract and Introduction, to make it clear that 3 generation family is split into two for these analyses.

(2) In the abstract, the statement “DNM burdens arising from parents of either sex who carried a single germline POLE or POLD1 mutation, or biallelic MUTYH mutations, were approximately 3- to 4-fold increased over controls.” is not correct.

This refers to the increased burden from a carrier parent, not total burden. We now provide greater clarity: “DNM burdens arising from a mother or father with a single germline POLE or POLD1 mutation, or biallelic MUTYH mutations, were approximately 3- to 4-fold increased over controls of the same sex”. We assume that the proportion of mutations phased to the mother or father can be used to estimate total DNMs from that parent, based on the strong correlation between phased and total DNMs that we show.

Based on the VAF distribution in Sup Fig 8, the mutations of MUTYH_C:II.1 (the biallelic MUTYH offspring) are likely post-zygotic and probably arose during clonal haematopoiesis. Therefore, they should not be considered de novo mutations. As such, based on the presented data, only offsprings with a POLE carrier parent have increased DNM (and more like 2-3 fold rather than 3-4 fold).

We do state that this patient was excluded from all of the main analyses and that the mutations are “apparent” DNMs.

(3) The bulk of the main results is summarised in Table 1. While this table is useful and informative, I feel that the description of the results would be easier to follow if some of the findings and comparisons could be presented in simple bargraphs with error bars and significance where relevant.

We would argue that some data are better shown in tables. To take a simple example, a relationship between parental age and DNM burden is messy to show in a simple chart, because the same parents have more than one child at different parental ages. We also note that we already use charts to show the results of the main analyses in which groups of patients and controls are compared (Supp. Figs. 3, 4, 7, 8 and 9 in the original version). Nevertheless, in the light of the reviewer's comments, we have brought these charts into the main manuscript (into a multi-panel Figure 1) in order to improve the data presentation.

(4) Line 142 "The number of phased SNVs correlated strongly with total DNMs (details not shown)." Related to the above comment, this can easily be shown as a scatterplot, perhaps as a supplementary figure.

We have now added a chart showing this with the regression equation.

(5) SBS38 is not particularly strong in MUTYH_CII.1. How many mutations are attributed to this signature? De novo signature extract should be performed to determine if this signature is readily detectable.

We now show the raw 96-channel spectrum and fitted spectrum, enabling the number of mutations ascribed to SBS38 to be determined. Much remains unknown here, such as the interaction between 5FU and MUTYH deficiency, and we do not wish to claim more than limited evidence of the effects of MUTYH deficiency.

We have performed de novo signature extraction in other contexts (e.g. cancer genomes), but were worried that this would be problematic in the current study, given the numbers of DNMs available. We have now performed de novo signature extraction across patient types, but unfortunately, in every set of mutations, we obtain only a single stable solution that is very similar to the underlying 96-channel spectrum.

(6) Similarly, for the bottom panel in Fig2, I don't think matching to ID signatures would be very reliable. For instance, although MUTYH_AII.1 has ~0.6 similarity to ID6, the sample only has 4 indels.

Indeed. We showed the DBS and INDEL signatures for completeness, but agree that they add little and are happy to remove them.

(7) Generally, the presentation of the mutational signatures is misleading as the signatures themselves are just taken from COSMIC with a % contribution assigned by SigProfiler. For the mutational signature analysis, the original trinucleotide mutational spectrum, along with the residue error after COSMIC signature fitting, should be provided. De novo extracted signatures should be shown.

The overriding reason for examining signatures was to check whether we specifically saw evidence of the expected mutational process in the DNMs from each family type. We contend that fitting the COSMIC signatures is a suitable, relatively sensitive method for this purpose, given the availability of good reference signatures, especially for *POLE* and *POLD1*. We have added back the removed the raw mutational spectra, which we wrongly thought to be of limited interest. (Please see above for de novo extraction response).

(8) It would be useful to include a supplementary table showing the chemotherapy that each patient may have been exposed to.

In the Methods, we now state that no parent had received chemotherapy prior to having children and that MUTYH_C:II.1 was the only child to have received chemotherapy.

(9) Panel labels are missing in Fig 2 and the figure legend of (c) doesn't seem to match what is shown in the figure.

The term "lower" was misleading and we apologise for this. It has been clarified in the new version of this figure.

(10) The results section could benefit from some subheadings to improve clarity.

Now added.

(11) Introduction line 91. I think a more commonly used term for biallelic MMR deficiency is constitutional MMR deficiency rather than congenital MMR

Thanks – this stupidly crept in by mistake and has been corrected.

(12) Line 96 "Families with germline POLE and POLD1 mutations provide a specific opportunity..." Provide a specific opportunity for what?

We have clarified this.

(13) Line 379, 382 Missing references

Thanks, now amended.

Jason Wong

Reviewer #3 (Remarks to the Author):

While germline mutations in DNA repair genes are known to contribute to increased mutagenesis in tumors, it remains an open question to what extent they contribute to de novo mutations in normal tissues from carriers and their offspring. In the current study, Sherwood and colleagues address this question by analyzing blood WGS from carriers of germline POLE, POLD1, and MUTYH mutations parents and their offspring from several independent families. They make several notable findings including that the overall increase in pre-zygotic DNMs is roughly 2- to 4-fold elevated over control families, but that this increase is almost entirely in non-coding regions of the genome. They also find that while DNMs are not elevated overall in POLD1 offspring, when the DNMs are phased to the parent there are detectable increases and this is affected by the sex of the carrier parent. They also make an interesting finding in post-zygotic DNMs in a MUTYH carrier who underwent 5FU treatment.

Robinson et al (Nature Genetics 2021) previously sequenced a number of normal tissues including intestinal crypts, endometrial glands, skin, sperm, and blood from fourteen POLE/POLD1 carriers. While the authors do cite this study on several occasions, this manuscript is written in such a way that a reader could be forgiven for not realizing that such POLE/D carrier normal tissue sequencing had been done. It strikes this reviewer as a curious missed opportunity to address some of the strengths of this study.

> We certainly do not wish to minimise the importance of Robinson et al – it is our own paper, after all! The two studies are complementary in that they address different aspects of the consequences of germline mutations in DNA repair genes. However, we are wary of making quantitative comparisons regarding mutation rates, given confounders such as unknown differences between the dynamics of germline and somatic tissues and the distinct tissue preparation and sequencing methods used for the somatic mutagenesis analysis.

Issues to be addressed:

The signature analysis was performed on pooled mutation data. The TCT mutations account for roughly $\sim 2/3$ (I apologize but the font is incredibly small on the y-axis label) of the 30% of total mutations attributed to SBS10a, meaning approximately $\sim 20\%$ of the total mutations in the pool are that POLE signature. Likewise SBS28 is 45% of 10% (4.5% total mutations). So the post-zygotic POLE-specific DNMs represent $\sim 25\%$ of total mutations when the total DNMs are increased 3-4-fold. An increase from 60 to 120 is a 60 DNM increase per individual meaning ~ 30 , or half of the increase, could be attributed to POLE. Could the authors speculate on what is driving the increase in non-POLE DNMs in the POLE samples?

> Signature extraction continues to evolve and become more refined. Some signatures are merged and others split into sub-signatures over time, and yet others identified as artefacts. Whist 10a, 10b and 28 probably represent the most active mutational processes in the specific samples from which they were identified (mostly colorectal and endometrial cancers), we are not confident that these three signatures capture the full spectrum of excess mutations caused by POLE mutations. Thus, whilst signature extraction methods seem to perform very well to detect presence/absence of POLE-specific signatures such as 10a or 10b, we are reluctant to make quantitative assessments such as the ones proposed by the reviewer in what is a small sample compared with cancer genome sequencing projects. Consequently, we do not exclude the possibility that the DNM excess in the germ line could all be down to POLE.

Supp Fig 3 should really indicate each data point. Supp Fig 3a also feels a bit misleading. The data demonstrate a clear difference between POLE and POLD1. The authors do show a difference in the phased mutations and discuss that the lack of difference may result from sex and size

sample biases. But the figure as presented implies differences in POLS relative to controls, when POLD1 is no different than controls.

> This has been changed as suggested. We have removed Supp. Fig. 3a.

I realize that in the legend, the author's state that the reasons are unclear, but this reviewer would be interested in hearing what is the authors' explanation for the SBS10d signature non-carrier parents (Supp Fig 6). They mention the low number of mutations and the low percentage (13%).

The POLD1 COSMIC signatures come from our own work on germline *POLD1* carriers (Robinson, 2021), and are not as well established as POLE signatures, owing to the much lower frequency of pathogenic *POLD1* mutations in sporadic tumours. The 10d signature in POLE non-carrier parents appears largely to come from one individual, the father (non-carrier) in the second part (generations 2 > 3) of family POLE_B. His DNMs (shown below) indicated SBS56, a signature closely related to 10d, but supposedly artefactual according to COSMIC. He contributed 29 of the 75 paternal non-carrier mutations in the POLE families and we can speculate that he was exposed environmentally to a source of C>A mutations. We have added this information to the supplementary figure legend.

Related to this is that SBS28 is at lower percentages in the POLE samples (10-11%). The total numbers of these mutations are not given but would perhaps be informative here.

The origins of SBS28 are not clear, and it may represent a mutational process that is not very active in the germ line. We now show raw 96-channel mutation spectra and fitted signatures for all groups.

The authors find increased DNMs only in phased mutations for *POLD1* and *MUTYH*. But only ~25% of DNMs were able to be phased. Do the authors think this is a technical limitation and that something (increased depth, etc) would identify more *POLD1* or *MUTYH* signatures? Or that there are simply very few pre-zygotic *POLD1*/*MUTYH* mutations to be identified? Or some other scenario?

The phasing is actually rather powerful in increasing signal:noise. Increased depth is very unlikely to help the phasing much, because the limiting factor is not enough parental SNPs to phase the DNMs. In fact, our study shows this, as it does have higher depth than others, but phased mutation proportions are raised very little. Small improvements to signatures could be derived from increased read depth, but this will still be limited by problems in distinguishing true

DNMs that have a low allele frequency caused by chance sampling of sequencing reads from background noise. Our strong suspicion is that there are few DNMs to be identified, but future studies using long read sequencing may succeed in phasing a larger proportion of DNMs.

Line 285 Missing period (".") between "extraction" and "This"

Thanks, now corrected.

REVIEWERS' COMMENTS

Reviewer #1 (Remarks to the Author):

The authors have adequately addressed all of the concerns brought up by this reviewer in the previous review. The newer additions to paper supplement add a bit more rigor the manuscript's analysis. Overall, the revisions are satisfactory and the reviewer has no additional major concerns.

A couple of minor text issues were noted:

- 1) Figure 1g is not cited anywhere in the main text
- 2) Lines 117 (Page 4) and 130 (Page 5) are redundant.

Reviewer #2 (Remarks to the Author):

Thank you for addressing my concerns. I have no further comments.

Reviewer #3 (Remarks to the Author):

The authors have been quite responsive to all three reviewers. This reviewer is satisfied with the responses and the manuscript in its current form.